# Novel Methods to Manipulate Autolysis in Sparkling Wine: Effects on Yeast

**DOI:** 10.3390/molecules26020387

**Published:** 2021-01-13

**Authors:** Gail B. Gnoinski, Simon A. Schmidt, Dugald C. Close, Karsten Goemann, Terry L. Pinfold, Fiona L. Kerslake

**Affiliations:** 1Horticulture Centre, Tasmanian Institute of Agriculture, University of Tasmania, Sandy Bay, Tasmania 7005, Australia; dugald.close@utas.edu.au (D.C.C.); fiona.kerslake@utas.edu.au (F.L.K.); 2The Australian Wine Research Institute, Glen Osmond, South Australia 5064, Australia; simon.schmidt@awri.com.au; 3Central Science Laboratory, University of Tasmania, Sandy Bay, Tasmania 7005, Australia; karsten.goemann@utas.edu.au; 4Tasmanian School of Medicine, University of Tasmania, Hobart, Tasmania 7000, Australia; terry.pinfold@utas.edu.au

**Keywords:** sparkling wine production, autolysis, microwave, ultrasound, β-glucanase enzymes, scanning electron microscopy, flow cytometry

## Abstract

Sparkling wine made by the traditional method (Méthode Traditionelle) develops a distinct and desirable flavour and aroma profile attributed to proteolytic processes during prolonged ageing on lees. Microwave, ultrasound and addition of β-glucanase enzymes were applied to accelerate the disruption of *Saccharomyces cerevisiae*, and added to the tirage solution for secondary fermentation in traditional sparkling winemaking. Scanning electron microscopy and flow cytometry analyses were used to observe and describe yeast whole-cell anatomy, and cell integrity and structure via propidium iodide (PI) permeability after 6-, 12- and 18-months post-tirage. Treatments applied produced features on lees that were distinct from that of the untreated control yeast. Whilst control yeast displayed budding cells (growth features) with smooth, cavitated and flat external cell appearances; microwave treated yeast cells exhibited modifications like ‘doughnut’ shapes immediately after treatment (time 0). Similar ‘doughnut’-shaped and ‘pitted/porous’ cell features were observed on progressively older lees from the control. Flow cytometry was used to discriminate yeast populations; features consistent with cell disruption were observed in the microwave, ultrasound and enzyme treatments, as evidenced by up to 4-fold increase in PI signal in the microwave treatment. Forward and side scatter signals reflected changes in size and structure of yeast cells, in all treatments applied. When flow cytometry was interpreted alongside the scanning electron microscopy images, bimodal populations of yeast cells with low and high PI intensities were revealed and distinctive ‘doughnut’-shaped cell features observed in association with the microwave treatment only at tirage, that were not observed until 12 months wine ageing in older lees from the control. This work offers both a rapid approach to visualise alterations to yeast cell surfaces and a better understanding of the mechanisms of yeast lysis. Microwave, ultrasound or β-glucanase enzymes are tools that could potentially initiate the release of yeast cell compounds into wine. Further investigation into the impact of such treatments on the flavour and aroma profiles of the wines through sensory evaluation is warranted.

## 1. Introduction

Sparkling wine made by the traditional method (*Méthode Traditionelle*) develops a distinct and desirable flavour and aroma profile, frequently described as ‘autolytic character’, that is attributed to proteolytic processes that occur during prolonged ageing in contact with yeast cells (lees) [1,2,3]. The traditional method of sparkling wine production encompasses two fermentation processes. Firstly, juice from pressed grapes undergoes a primary fermentation to produce a base wine. Secondly, this base wine receives a subsequent addition of yeast and sugar (*liqueur de tirage*), often combined with an adjuvant mixture comprised of bentonite, alginate and nutrients, which initiates secondary fermentation. The process occurs in closed bottles in which yeast consume sugar and this process produces ethanol and carbon dioxide (about six standard atmospheres) that result in effervescence. Secondary fermentation is generally complete in two to three months after which lengthy maturation in contact with yeast is considered necessary to develop sensorial complexity [1,4,5,6,7].

Autolysis begins between two to four months post-tirage [1,6]. Yeast starvation induced by a lack of nutrients, the presence of carbon dioxide, high alcohol concentration, low pH (3.0–3.5) and low temperature (15 °C) storage conditions trigger autophagy [5,8]. Autophagy is a catabolic process involving the degradation of macromolecules inside the vacuole or lysosome, proposed as an activator of the autolytic process [9]. Internal proteases act to hydrolyse lysosomal and cytoplasmic membranes as autolysis progresses. Irreversible degradation of cell wall components like glucans and mannoproteins increases cell wall porosity facilitating the release of degraded constituents into the wine. The slow rate of enzymatic activity delays the autolytic process and consequently wines are left in contact with lees for several months or years to benefit from positive autolytic effects.

Yeast autolysis modifies sparkling wine flavour and aroma profiles compared to their base wines [5,10]. The changes in wine texture and composition during ageing have been attributed to an increase in yeast derived components in the finished wine [11,12]. These yeast derived compounds typically include cytoplasmic (amino acids, peptides, fatty acids, nucleotides) and cell wall material (proteins, mannoproteins and polysaccharides).

Methods to induce autolysis and facilitate the development of aged wine character in sparkling wine have been approached in a multitude of ways. Perpetuini et al. [13] found a wide diversity of autolytic potential in *S. cerevisiae* yeast strains, and suggested this biodiversity could be employed to select new starter strains to improve sparkling wine production. The use of combinations of killer toxin positive and negative strains has been proposed as a means of promoting autolysis during secondary fermentation [6], indeed showing increases in total protein concentration, used as a marker of autolysis, when such combinations were employed in laboratory trials. Lombardi et al. [14] demonstrated that killer and sensitive *S. cerevisiae* and *S. bayanus* strain combinations influenced free amino acids, total protein and polysaccharide concentrations in pilot scale production of sparkling wine, following an ageing period of three months, whereas sparkling wines produced without the use of these yeasts, attained equivalent concentrations after nine months. Rather than attempting to induce autolysis, La Gatta et al. [15] attempted to augment the tirage solution with spent lees recovered following production of base wine. These authors showed increased proteolysis and decreased peptide concentrations associated with base wine lees addition, resulting in positive effects on wine aromatic characters but detrimental impacts on foam stability [15].

Various physical interventions have been trialled in attempts to directly disrupt yeast cell integrity. *S. bayanus* subjected to high pressure homogenisation was evaluated for its potential to release cell constituents including proteins and amino acids, and the impact of its addition during the production of still wine reported [16]. The treatment resulted in lower ester concentrations and the release of soluble macromolecules but overall limited impact on wine sensory attributes was demonstrated [17]. The application of microwave or ultrasound energy to yeast have been proposed to assist in the mechanical breakdown of cells. Microwave heating was used to rupture the cell matrix and extract compounds from lysed cells [18]. Ultrasound and microwave treatments of lees showed great efficiency in increasing the release of polysaccharides as well as compounds like acetaldehyde and vitisin B, that may help better preserve the colour of *Vitis vinifera* L. cv. Tempranillo wine over time [19]. 

Enzyme activity of yeast is generally inhibited under oenological conditions of low pH, low temperature and high alcohol content, therefore use of exogenous enzyme preparations may overcome the problem associated with low activity of yeast endogenous enzymes [20]. The addition of enzyme preparations rich in β-glucanase have been used to hydrolyse yeast cell walls and release cells contents into the wine, altering aged characters and enhancing the anti-oxidant properties of sparkling wine [21,22,23,24]. The addition of exogenous 1,3-β-glucanase has also been used to promote the release of mannoproteins from the cell wall [25,26]. However, the impact of enzyme treatment on the release of imbedded cell molecules like lipids and intracellular compounds such as nucleotides is less well understood [27]. The microwave, ultrasound and β-glucanase enzyme treatments described above have not been examined sufficiently to determine the longer-term effects on wine quality.

Yeast cells have been shown to exhibit elongated and ovoid morphologies during fermentation, compared to the more spherical shaped cells in sparkling wine lees aged for 12 months [28]. In contrast, morphologies associated with cellular degradation appeared as a reduction in cell volume during induced autolysis in a model wine whereas cells surface wrinkles/folds were more apparent on yeast cells aged in wine for 12 months. These features were attributed to plasmolysis and were not found on cells isolated during fermentation [29]. Transmission electron microscopy studies of yeast lees following extended sparkling wine aging showed that despite evident thinning and degradation, the cell wall remained unbroken after 48 months wine ageing [30]. Cell membrane breakage was directly observed as was morphological evidence of autophagy. Structural changes to *S. cerevisiae* have been observed over lengthy aging on lees (up to 19 years) post-secondary fermentation [31]. These studies also demonstrated the resilience of the yeast cell wall despite the breakdown of internal membranes.

In the work reported here, yeast morphology and cell integrity and structure during sparkling wine maturation was investigated, in isolation from oenological impacts on wines, which warrants further investigation. The morphological effects of microwave, ultrasound and β-glucanase enzyme treatments of *S. cerevisiae* prior to their addition as a component of a tirage solution, and after being disgorged from sparkling wine, were evaluated qualitatively by visualisation using scanning electron microscopy, and compared to a control sparkling wine. Lees samples were further examined by flow cytometry analyses to quantify yeast cell impacts, with a combination of fluorescent propidium iodide (PI) and flow cytometry used to measure cell integrity. Flow cytometry analyses of PI uptake is a well-established and rapid method for monitoring cell death and is used on the basis that the intact membrane of viable cells excludes PI and that loss of this permeability barrier represents irreparable damage and cell death. In non-viable cells the PI can ingress into the cell and bind to nucleic acid resulting in red fluorescence [32]. Furthermore, flow cytometry analyses can be used to infer conditions of the physical state of cells. When incident light hits a cell the light scatter is related to cell mass, structure, surface properties and the optical density of internal cell contents [33]. Parameters like forward scatter (FSC) primarily indicate cell size and shape, and side scatter (SSC) indicates the internal granularity of a cell and surface roughness. The interactions of these parameters provide information on the composition of the cell. 

## 2. Results

### 2.1. Scanning Electron Microscopy Observation of Yeast Cells

Table 1 defines the set of visual features used to characterise yeast whole-cell surface features. Example images showing the types of morphologies observed in *S. cerevisiae* cells and categorised as per Table 1 are shown in Figure 1. The six categories are: ‘smooth’, ‘cavitated’, ‘flat’, ‘pitted/porous’, ‘doughnut’-shaped or ‘fragments’. These visual feature categories were derived from a visual review of all images associated with the study.

Three morphological features characterised the control yeast at tirage (time 0), being smooth, cavitated and flat (Figure 2). After 6 months in bottle there was almost a complete loss of smooth cells, an increase in the proportion of cells exhibiting cavitated and the appearance of ‘doughnut’- shaped cells (Figure 2a). By 12 months the proportion of cells showing cavitated-like features exceeded 90% and this was associated with the loss of flat features and the appearance of pitted cells and cells fragments. The morphological profile did not substantially change between 12- and 18-months.

Comparisons between treated cells to untreated cells at time 0 revealed that the proportion of smooth cells in microwave (18 ± 3%) and ultrasound (9 ± 1%) treatments were significantly less than the control (39 ± 7%) (Figure 2b). However, there was no significant difference in the proportion of ‘smooth’ cells in the β-glucanase treated cells (34 ± 5%) relative to the control. Cavitated features were observed more frequently in ultrasound (78 ± 9%), microwave (65 ± 9%) and β-glucanase (57 ± 3%) relative to the control (34 ± 5%) at time 0. ‘Doughnut’-shaped cells occurred only in the microwave treated (1 ± 2%) yeast at time 0. Flat cells were found in all treatments but were predominantly present in untreated cells (26 ± 6%), and to a lesser extent in the microwave (18 ± 5%), ultrasound (12 ± 6%) and β-glucanase enzyme treatments (8 ± 3%). Fragments were present in all treatments, and there was no significant difference between the control and the treatments. Pitted/porous cells were not observed at time 0 in any of the treatments.

Overall, morphological features observed on the microwave and ultrasound treated cells at time 0 were similar to these found on control lees after six months ageing of sparkling wine (Figure 2). More smooth cells were found on the control lees compared to the treatments at time 0, however, the microwave and ultrasound treated cells exhibited more smooth cells at time 0 compared to the control cells at 6 months (Figure 2b).

### 2.2. Flow Cytometry Analysis

At time 0, microwave treated cells had a 4-fold higher PIF (Figure 3a) than the control population (*p* = 0.015). This can also be seen as a right-shift along the *x*-axis of Figure 4b for the microwave treated cells relative to fresh yeast (labelled control, shaded). There were no significant differences in PI intensities for the ultrasound (*p* = 0.862) or β-glucanase enzyme (*p* = 0.811) treatments compared to the control, however these treatments displayed two-fold shifts to higher PI intensities (Figure 4c–d). There was little background fluorescence observed in this work indicated by the absence of signal in unlabelled fresh yeast (Figure 4).

The effect of ageing on PI staining potential is demonstrated by an analysis of the control wines over time (Figure 3b). The control exhibited a significant decrease (*p* = 0.017) in PI fluorescence intensities at 6-, 12- and 18-months relative to time 0 (about 4200 au). PI fluorescence intensities were found to decrease by 2-fold at 6 months (about 1954 au) and 12-month bottle age (about 1951 au), and by 3-fold at 18 months (1378 au). This is also demonstrated by the shift to the left (to lower PI intensities) along the *x*-axis in Figure 5a–d.

There was a bimodal distribution of PI fluorescence from yeast analysed at time 0 (Figure 6a,b), forming population 1 of lower PI, low SSC (ultrasound treatment mainly, and less so from the control and β-glucanase enzyme treatment); and population 2 at higher PI, variable SSC (from the control on the lower PI end of population 2, the ultrasound treatment and β-glucanase enzyme treatment had similar PI signals, with the microwave treatment on the high PI end of population 2).

The small cluster of cells at the lower PI spectrum (population 1) originated from ultrasound treated yeast (black) and yeast from the β-glucanase enzymes (grey) treatment, and also the control (magenta) (Figure 6a,b). 

Analysis of the control lees over time, indicated increased SSC signal from 6- and 12-months bottle age, and FSC signal exhibited less variation (Figure 7a,b).

## 3. Discussion

This work investigated the impact of treatments designed to disrupt yeast cell integrity and used to supplement the tirage solution in the secondary fermentation stage of sparkling wine production. Microwave, ultrasound and β-glucanase enzyme treatments had various and distinct impacts on yeast physical structure as assessed by scanning-electron microscopy and flow cytometry.

### 3.1. Morphological Features of Yeast Cells

In this study, we found typical yeast morphological features of smooth-surfaced, ovoid and oval in shape with a turgid appearance [34] in addition to budding from (initially) continued reproduction. In addition, we found altered morphologies of cavitated, pitted and porous cell surfaces, flat, ‘doughnut’-shapes and fragmented yeast. These observations enabled systematic categorisation and quantification of yeast morphologies associated with the aging of sparkling wine, as well as feature profiles distinct to the novel treatments of this study. Previous studies of yeast during autolysis in sparkling winemaking have described progressive degradation of internal cell structure [30,31], involving the dissolution of the stratified structure of the cell wall although, despite thinning and degradation, the cell wall has been shown to remain unbroken throughout wine ageing. Further, Piton et al. [31] described transformation of the outer cell wall layer by removal of polysaccharide (mannans) compounds that give the cell its shape and rigidity. Garcia Martin et al. [35] described features of ‘wrinkles and folds’ that may lead to the cavitated features found in this study, though we are not aware of previous reports of flat, pitted/porous cell features or ‘doughnut’-shaped morphologies arising during the process of autolysis in traditional sparkling wine production.

### 3.2. Changes in Yeast Morphological Features During Secondary Fermentation with aging of Sparkling Wine

During autolysis, in the absence of novel treatments, yeast morphology clearly shifted from a high proportion of smooth cells (about 40%) at time 0 to a negligible proportion from 6 months onwards. Cavitated cells steadily increased in proportion from 30% at time 0 to almost 100% at 18 months, whilst in contrast yeast cells of flat appearance steadily decreased in proportion from about 26% to about 2% at 18 months. Pitted/porous and ‘doughnut’-shaped cells increased in frequency at 12- to 18-months. Consistent with our findings of cavitated cells at age tirage (time 0), Piton, et al. [31] reported changes to the yeast cell structure as early as 6 weeks after bottling that were reflected in neutral lipid accumulation at the expense of polar lipids (cell wall constituents). In contrast modification to the yeast cell wall during aging of champagne began at 6 months, with the removal of the inner cell layer even when cells were plasmolysed [31]. Decreased polysaccharide content and amino acid loss from yeast starvation trigger cell derived β-1,3 glucanases to degrade the cell wall causing collapse [36] with these endogenous glucanases present in the cell wall up to 4 months after cell death [37]. Our observations indicate that the degradation processes continued to impact cellular structures up to age 18 months in a continuous way given the steady increase in cavitated, and commensurate decrease of flat-appearing, yeast cells.

### 3.3. Effects of Novel Treatments on Yeast Morphological Features

At time 0, the proportion of visually smooth cells were observed in only 18% and 9% of the microwave and ultrasound treatment cells, respectively, relative to 40% of the control and 34% of the β-glucanase enzyme treatment cells. The proportion of cavitated cells in the control was 34% compared to 57%, 65% and 78% in the β-glucanase enzyme, microwave and ultrasound treatments, respectively. Taken together these findings indicate that these technologies effectively disrupted yeast cell integrity, given that the extent of the cavitated features were similar to that observed in control treatment wines at 6- and 12- months of bottle age. Fast and efficient extraction using microwaves resulted in higher yield of phenolic compounds in plant based foods [18]. Garcia Martin et al. [35] evaluated the effects of ultrasound assisted lysis by analysing the release of proteins and polysaccharides into model wine, and the viability of cells contained in the lees and found complete inactivation of cells from ultrasound at the point of maximum protein release into the model wine. Clearly, the treatments trialled in this study had significant impacts on yeast cellular morphologies with differences between the treatments pointing to different mechanisms of disruption.

Of note, is the distinctive ‘doughnut’-shaped cell features observed in association with the microwave treatment only at time 0. The ‘doughnut’-shaped feature is suggested to represent a severe collapse or a ‘hole’ in the *S. cerevisiae* cell wall. Although not reported in wine yeasts, ‘doughnut’-shaped appearance of yeasts has been reported when ionic liquids were applied as a pre-treatment in biofuel technology and cell protein biomass production processes [38,39,40]. ‘Holed’ appearance [38] and invaginations from protoplast conversion from removal of the cell wall by treatment with lyticase, a complex of endoglucanase and protease, has been reported [39]. We propose a mechanism whereby the microwave treatment severely disrupt the yeast cell wall via heat transfer that breaks hydrogen bonds between glucosidic monomers in the cellulose matrix of the cell wall.

Consistent with our findings of two-fold more cavitated features of cells treated by ultrasound, relative to the control, Garcia Martin et al. [35] reported that ultrasound-assisted lysis of light lees was associated with degradation of cell walls and altered morphological appearance described as ‘wrinkles’, that were attributed to plasmolysis. In contrast to the reports of Garcia Martin et al. [35] we did not find long ridges emerging from cell walls. However, the light lees used in their study had undergone autolysis during previous barrel ageing, in contrast to our relatively young yeast cells.

In contrast to the observations associated with the microwave and ultrasound treatments, the β-glucanase enzyme treatment had a similar proportion of smooth yeast cells at time 0 relative to the control treatment. However, Palermo et al. [26] reported β-glucanase enzyme-induced release of polysaccharides in two to three weeks in model wine, relative to five months in conventional autolysis. Our findings indicate a slower process, relative to microwave and ultrasound, but consistent with the findings of Palomero et al. [26] and others [22,23] we found high levels of cavitated β-glucanase enzyme treated cells, relative to the control treatment, that may be indicative of release of polysaccharides arising from cell wall disruption. A wide diversity of autolytic potential in *S. cerevisiae* yeast strains was reported by Perpetuini et al. [13], that could be employed to improve sparkling wine quality. It is suggested the morphological impacts of yeast treatments reported here could likely be similar in other strains of yeast used for sparkling wine production.

### 3.4. Yeast Cell Integrity as Measured by PI Permeability

At time 0, the microwave treated yeast had a 4-fold higher PIF intensity (16,858 au) relative to the control (4217 au). The shift to elevated uptake of propidium iodide by microwave treated yeast suggests more disruption relative to the ultrasound treatment or the β-glucanase enzyme treatments. This was consistent with observations of greater morphological disruption to yeast in the microwave, relative to other treatments. Similarly, Guzzon and Larcher [41] reported an intense signal in the red fluorescence PI channel following heat treatment of active dry yeast at 80 °C for 10 min in a water bath to kill all vegetative forms. Further to that the increase in red fluorescence of PI was inversely proportional to *S. cerevisiae* viability following acetic acid treatment (10% *v*/*v*, 10 min at 25 °C) [31]. 

There was a decrease in propidium iodide fluorescence intensities for control yeast disgorged at 6-, 12- and 18- months relative to control yeast at tirage. The shift to lower PI intensities (2-fold drop) with increasing wine age is consistent with a reduction in nucleic acid in control yeast, potentially due to less cell degradation compared to the treated cells, and less rupture or disorganisation of the cell internal structure. The trend is consistent with flow cytometry analyses reported by Guzzon and Larcher [41] who monitored the evolution of *S. cerevisiae* strains FEM111 and FEM 222 during sparkling wine production using flow cytometry analyses at bottling through to the end of alcoholic fermentation until stabilization of the pressure at five atmospheres inside the bottle. They identified populations of live cells, dead cells and a third population of events corresponding to ‘compromised/damaged’ cells that were biologically active while at the same time exhibiting impaired cell membrane permeability [41]. We monitored secondary fermentation in the bottle over a maturation period of 18 months, and found loss of nucleic acid from the cells over that time with much of the PI interactive material lost within the first 6 months. Decreased nucleic acid content, possibly due to degradation, is a potential consequence of increased cell permeability over time. It has previously been observed that natural autolysis is a slow process during ageing of sparkling wines produced by the traditional method [4] and that loss of cytoplasmic contents is a common feature of extended maturation [28].

The interactions of flow cytometry side scatter (SSC) and forward scatter (FSC) parameters together with PI intensities, elucidated possible causes of changes to the control yeast cells over time [42]. We found greater complexity and a change in the control yeast cell structure (internal granularity or surface roughness) indicated by the SSC signal at 12 months bottle age. SEM observations indicated elevated levels of pitted/porous features on control yeast cells at the 12 month stage and a significant increase in the proportion of ‘doughnut’-shaped cells at 12 months. These observations are consistent with cell damage that could lead to a reduction in nucleic acid from the control yeast, and is a possible explanation for the decrease in PI intensities measured in the control yeast over time. Forward scatter signal characteristics indicated a small distribution in cell size that was further supported by SEM images that showed a minor cell size variation (3 to 5 µm) in the control wine lees at 0-, 6-, 12- and 18-months bottle age.

Interactions of PI fluorescence, SSC and FSC parameters indicated a bimodal distribution, forming two clusters of cells, in the control and the treated yeast at tirage (time 0). These clusters could be described as population 1 of lower PI, low SSC (from the ultrasound treatment mainly, and less from the control and the β-glucanase enzyme treatment); and population 2 at higher PI, variable SSC (from the control on the lower PI end of population, the ultrasound treatment and β-glucanase enzyme treatment have similar PI signals, with the microwave treatment on the high PI end of population 2). These results show distinct effects of microwave, ultrasound or β-glucanase enzyme treatments that are of greater precision than solely morphological observation, thus demonstrating the benefit of using both techniques in a complementary approach. The focus of the study was on the treatment effects on yeast morphology, however, results indicate further investigation of oenological impacts on wines is warranted.

## 4. Materials and Methods 

### 4.1. Sparkling Wine Production

#### 4.1.1. Yeast Culture

A commercial yeast used for sparkling winemaking, *Saccharomyces cerevisiae* IOC 18-2007 (Institut Oenologique de Champagne, Lallemand, France) was used to prepare a yeast culture for the *liqueur de tirage* for secondary fermentation for bottle-fermented sparkling wine. The protocol for preparing the yeast culture comprised three steps (a) rehydration, (b) gradual acclimation to the alcoholic medium and (c) preparation of the starter culture in the active growth phase [43]. Yeast was prepared according to the manufacturer’s recommendations at room temperature (25 °C) for about 30 min and adapted to growth in the base wine (Pied de Cuve, Comite Interprofessionel du vin de Champagne). Nutrients of 0.1 g L^−1^ diammonium phosphate, 0.04 g L^−1^ Cerivit (Lallemand, France) and 23 g L^−1^ sugar, together with base wine were added to the aerated culture to prepare the *liqueur de tirage* and propagated to 2 × 10^6^ cells mL^−1^. The yeast culture was chilled to 4 °C to terminate cell growth and kept cold until required.

#### 4.1.2. Yeast Treatments

Three separate yeast cultures propagated to 2 × 10^6^ cells mL^−1^ were prepared for use in three treatments for addition into the tirage solution. The first treatment comprised treating the yeast with microwave. A domestic microwave oven (Panasonic ‘the Genius’ 1100 W, 50 Hz, Shanghai, China) was operated at full power and the yeast culture was heated in a 1.0 L Schott bottle for 90 s in three cycles, heating the yeast culture up to 99 °C.

For the ultrasound treatment, sonication was carried out using an ultrasonic bath (Soniclean 2000TD Ultrasonic Bath, Australia). The bath was operated at a constant frequency 50 kHz, with an electrical power of 350 W. A Schott bottle of yeast (1.0 L) was closed and placed in a fixed position in the water bath for five cycles of 15 min each, with circulating cold water to maintain the water temperature in the bath between 20 °C to 25 °C. Cacciola et al. [44] found that ultrasound treatment duration rather than its intensity gave significant release of colloids from yeast (polysaccharides and glycoproteins) in a model wine solution.

The enzyme preparation used for the β-glucanase enzyme treatment was commercially available VinoTaste^®^ Pro, (Novozymes, Bagsvaerd, Denmark), containing 1 to 5 weight % β-glucanase active enzyme protein and 1 to 5 weight % polygalacturonase active enzyme protein. The enzyme activity 2500 PGNU g^−1^, as per the manufacturer’s product data sheet. A Schott bottle was filled with 1.0 L yeast and 5 g L^−1^ β-glucanase enzymes was added, mixed well, closed and maintained at 20 °C for 24 h. 

#### 4.1.3. Preparation of Sparkling Wines

A commercially produced 2016 vintage base wine blend of *Vitis vinifera* L. cv. Chardonnay and Pinot noir (pH 3.18, titratable acidity (TA) 6.0 g L^−1^, alcohol 11.1% *v*/*v*, free SO_2_ 33 ppm, total SO_2_ 133 ppm) from a blend of sub-regions of Tasmania, Australia was used for this trial (Figure 8). Standard 750 mL sparkling wine bottles were filled with the Chardonnay/Pinot noir base wine blend and 22.5 mL of *liqueur de tirage* was added. Adjuvant was not included in the *liqueur de tirage*, to facilitate better visualisation of the lees. For the yeast treatment wines, 7.5 mL of each yeast treatment (microwave or ultrasound or enzyme) was further added to each bottle corresponding to that treatment. There were four replicates (n = 4) for each for the three treatments plus the control (n = 4) and wine were produced for sampling at three time points, namely, 6-, 12- and 18 months (total = 48). The bottles were capped with crown seals and positioned horizontally at standard tirage conditions (about 15 °C), for the secondary fermentation and aged on lees for up to 18 months.

Disgorgement of lees from four bottles of sparkling wine per treatment plus the control occurred after 6-, 12- and 18-months. Sparkling wine bottles were riddled to a vertical position by turning twice a week over an eight week period leading up to disgorgement. Following that, the wines were taken from storage (15 °C) and cooled overnight to 4 °C. A slurry was prepared from salt (NaCl) and water, in an upright chest freezer, to freeze the bottle necks containing the lees. To avoid cross contamination, an enclosed plastic drum was thoroughly sanitised, including a final rinse with a potassium metabisulphite (PMS) solution prior to collecting lees and between disgorging each wine. Lees was recovered by disgorging each sparkling wine bottle individually into the enclosed plastic drum. Bottles of sparkling wine were closed with crown seals without addition of a *liqueur de expedition.* Disgorged lees was not subjected to centrifugation and was stored overnight at 4 °C to settle from wine prior to scanning electron microscopy and flow cytometry analyses.

### 4.2. Scanning Electron Microscopy Analyses

Yeast cell morphology was examined on recovered lees from disgorgement after 6-, 12- and 18- months ageing, using the FEI MLA 650 scanning electron microscope (FEI Company, Hillsboro, OR, USA) in environmental (ESEM) mode at the Central Science Laboratory, University of Tasmania. ESEM has been widely used to visualise biological samples [45] and provide high resolution images of cell surfaces at low vacuum settings. Cells were not dehydrated or coated with conductive material for SEM imaging, in order to minimise introduction of damage to cells.

Approximately 1.5 mL of the disgorged lees in residual liquid (from sparkling wine) were transferred to 2.5 mL centrifuge tubes and allowed to settle. Approximately 0.1 mL of the lees containing liquid was then transferred onto a 9 mm diameter flat, shallow, steel dish using a needle syringe, placed on a Peltier cooling stage, and held at 5 °C temperature during ESEM imaging. Two water vapour purging cycles from 500 to 1300 Pa pressure were carried out during initial pump down of the specimen chamber, afterwards the water vapour pressure was held at 550 Pa for imaging. Images were acquired at 5 kV accelerating voltage, 10 picoamps beam current, and a working distance of 5 to 6 mm. Whole-cell surface features of *S. cerevisiae* were visualised by ESEM, and the features were classified into categories using image analysis.

### 4.3. Image Analyses

A set of SEM images depicting yeast cells from disgorged lees were collected at tirage (time 0) and after ageing on lees at 6-, 12- and 18-months bottle age. ESEM image textures are complex, and a machine learning technique or similar for the classification of wine lees which relies less on human decision making was outside of the scope of this project. Therefore, semi-automated processing of the SEM images was performed using ImageJ/Fiji software [46,47,48]. The ImageJ plug-ins “Cell Counter” was used to tally whole-cell surfaces visually, and features were classified into categories and assigned descriptions such as ‘smooth’, ‘cavitated’, ‘flat’, ‘pitted/porous’, ‘doughnut’-shaped and ‘fragment’ in the SEM images. 

Four images (replicates) each the control and the three yeast treatments collected at time 0 were processed, together with eight images each of control lees collected after 6-, 12-, and 18- months were processed, to account for sample variability. Each processed image contained about 40–150 cells that were classified, giving about 80 cells on average per image. There was a high degree of concordance between images of the same treatment and treatment replicates.

### 4.4. Flow Cytometry Analyses

Flow cytometry was used to assess cell permeability and nucleic acid content of *S. cerevisiae* cells based on the uptake of propidium iodide (PI), a fluorescent nucleic acid stain (excitation 488 nm and emission 575 nm), that is excluded from cells with intact plasma membranes [42].

Lees collected from four 750 mL sparkling wine bottles (replicates) each for control and for the three treatments at time 0 and after 6-, 12- and 18-months post-disgorgement, were evaluated for cell membrane permeability, via propidium iodide fluorescence (PIF) and flow cytometry (FCM) with the median fluorescence intensity (MFI) of each sample recorded [41]. For each of the analyses, 10 μL yeast of sample was combined with 200 μL of phosphate-buffered saline (PBS) and 1 μL propidium iodide (PI, 2 mg/L solution in water, Sigma Aldrich, St Louis, MO, USA) for 10 min. Prior to FCM analyses, samples were homogenised by gentle vortex for two seconds. Samples were then analysed by a BD FACS Canto™ II flow cytometer (Becton Dickinson Immunocytometry Systems, San Jose, CA, USA) equipped with an air-cooled 488 nm solid state, 20 mW laser with the standard filter setup. The flow cytometer quality control was checked using BD™ Cytometer Setup and Tracking Beads. Freshly prepared *S. cerevisiae* yeast, according to the method described in Section 4.1.1, was analysed without PI staining (unlabelled control) with each batch of samples at each disgorging time point, after 6-, 12- and 18-months wine ageing. The FSC and SSC parameters (channels) of the FCM instrument were adjusted using the unlabelled *S. cerevisiae* before sample analyses. The FSC signal was used as the trigger signal. The FCS and SSC voltages were adjusted to place the yeast on a linear scale within a bivariant plot of FSC vs. SSC with suitable threshold set to eliminate smaller particles and noise from the collected data.

Plots of FSC-height vs. FSC-area were used to discriminate coincident events from the analysis. The red fluorescence of the PI was detected through a 670 nm long-pass filter [41]. For all parameters, both area and height readings were recorded on a log scale, and PI fluorescence intensity peak height (PI-H) is depicted in Figure 4 and Figure 5. CST voltages for the detector were chosen to ensure optimal sensitivity of the instrument. To minimise cell coincidence and reduce the coefficient of variation (CV), the flow rate was set to the lowest rate for tube acquisition (nominally 1 μLs^−1^). A minimum of 10,000 events were recorded per sample. Post-acquisition analysis was performed using FCS Express version 6 flow cytometry software (De Novo Software, Pasadena, CA, USA). A gating strategy together with PI/FSC/SSC histograms and dot plots were used to identify the yeast cell population in a sample representative of 1.5 mL sample of disgorged yeast.

### 4.5. Statistical Analyses

Quantitative data expressed as means ± standard deviations were presented using GraphPad Prism software version 8.0 for MAC, GraphPad Software, San Diego, CA, USA, www.graphpad.com. The data were evaluated with one-way analysis of variance (ANOVA) and comparison between treatments was analysed using Dunnett’s multiple comparison tests. The differences were considered significant when *p* ≤ 0.05 (*), *p* ≤ 0.01 (**), *p* ≤ 0.001 (***) and *p* ≤ 0.0001 (****) or not significant (ns).

## 5. Conclusions

The addition of yeast treatments, that have the potential for earlier release of cellular components, to the tirage solution could reduce the time required on lees to achieve the characteristics of mature sparkling wine. This study demonstrated that treatment of yeast with microwave, ultrasound or β-glucanase enzymes, prior to addition to a *liqueur de tirage,* caused morphological disruptions that were associated with differences in yeast permeability. This disruption was indicated by propidium iodide permeable yeast cell membranes. Modifications to cell structure from the microwave treatment produced unique ‘doughnut’-shaped features and a range of morphological effects to yeast at time of addition (tirage or time 0) that were similar to older lees from the control after 6-, 12- and 18-months ageing. These results indicate that microwave, ultrasound or β-glucanase enzymes are tools that could potentially initiate the release of yeast cell compounds into wine, and further investigation is warranted into the impact of such treatments on the flavour and aroma profiles of the wines through sensory evaluation.

## Figures and Tables

**Figure 1 molecules-26-00387-f001:**
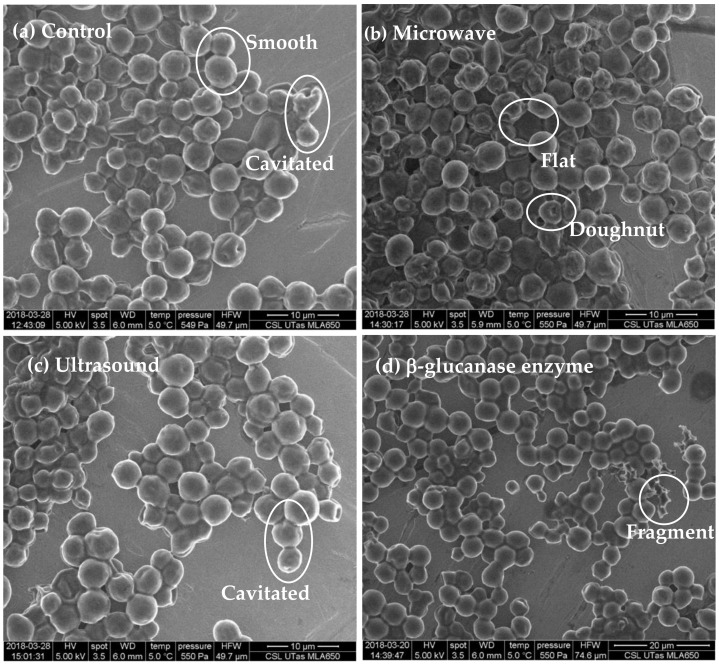
Scanning electron microscope images of *S. cerevisiae* whole-cells at tirage (time 0) exhibiting surface features of: (**a**) Control (untreated) cells exhibit ‘smooth’ external surfaces, budding, mother-daughter cells and ‘cavitated’ features on some cells; (**b**) Microwave treated cells displaying ‘flat’ and ‘doughnut’-shaped features; (**c**) Ultrasound treated cell surfaces displaying ‘cavitated’ features and (**d**) β-glucanase enzyme treated cells showing ‘cavitated’ features and ‘fragments’.

**Figure 2 molecules-26-00387-f002:**
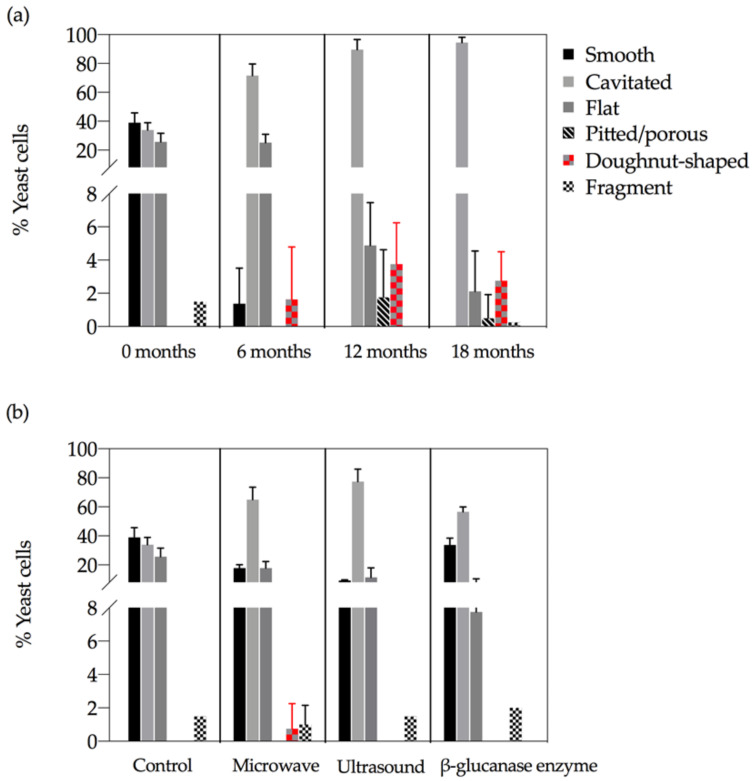
(**a**) Development of surface features over time on *S. cerevisiae* whole-cells in the control, and (**b**) Proportions of surface features observed in the control and the treatments at tirage (time 0).

**Figure 3 molecules-26-00387-f003:**
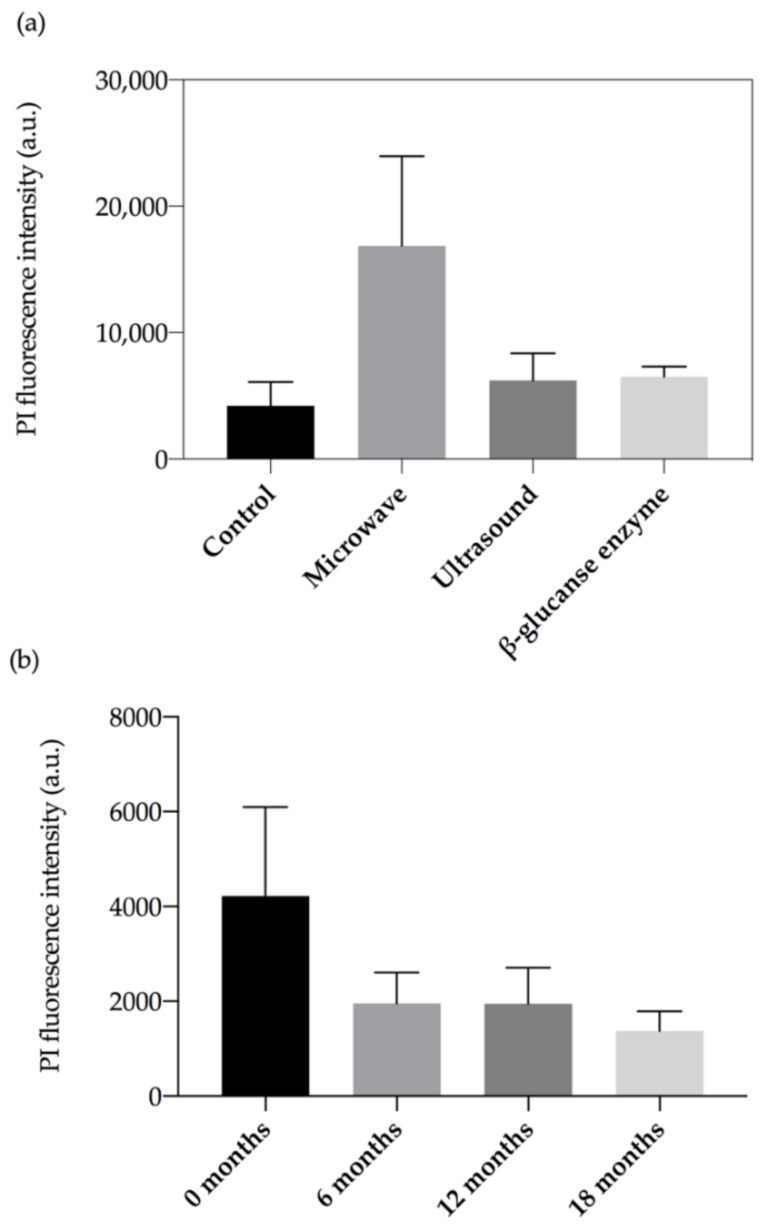
Median fluorescence intensities (MFI) of PI stained *S. cerevisiae* cells, n = 4 and error bars show standard deviation, depicting (**a**) Comparison of different treatments at tirage (time 0), and (**b**) Untreated control lees matured in bottle for different lengths of time.

**Figure 4 molecules-26-00387-f004:**
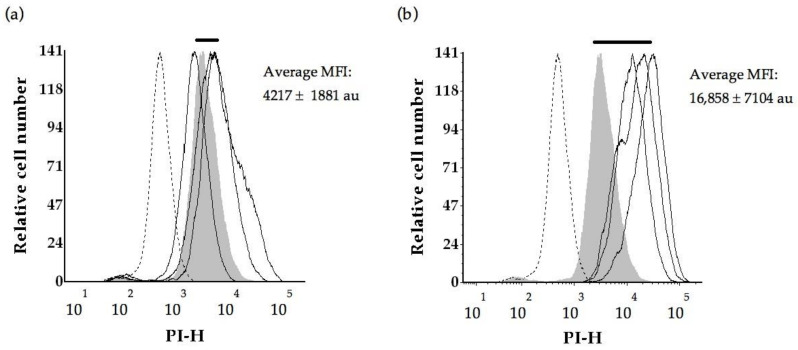
Propidium iodide fluorescence intensities measured for yeast cells at tirage (time 0). Solid bar represents the shift in PI intensities with respect to the labelled control, dotted line represents the unlabelled control, shaded area represents the labelled control, solid lines represent yeast replicates from (**a**) the control, (**b**) microwave treatment, (**c**) ultrasound treatment and (**d**) β-glucanase enzyme treatment. MFI denotes median fluorescence intensity of PI stained *S. cerevisiae* cells and PI-H denotes propidium iodide fluorescence peak height.

**Figure 5 molecules-26-00387-f005:**
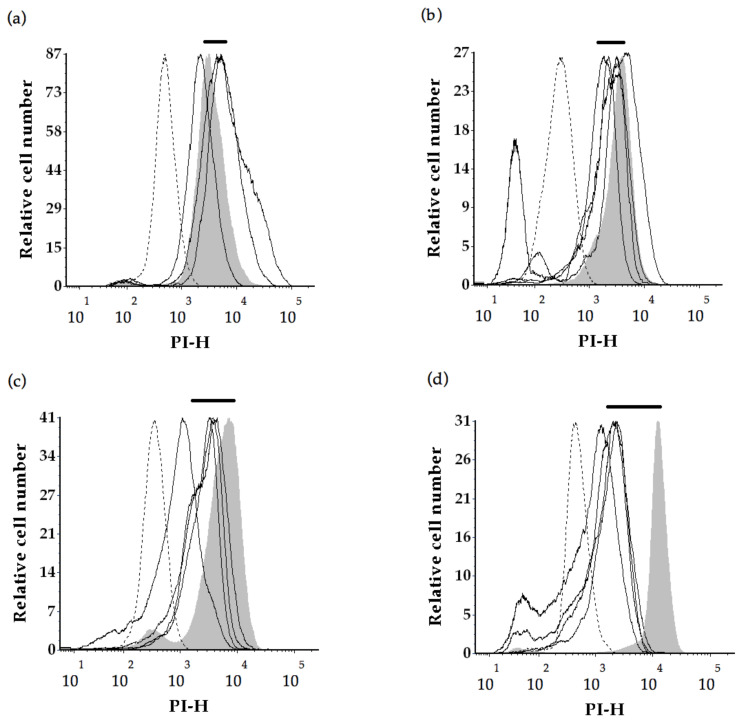
Fluorescence intensities (PI-H) measured for yeast cells from the control lees disgorged after (**a**) 0 months, (**b**) 6 months, (**c**) 12 months, and (**d**) 18 months ageing. Solid bar represents the shift in PI intensities with respect to the labelled control, dotted lines represent the unlabelled control, shaded areas represent labelled control, solid lines represent yeast replicates. PI-H denotes propidium iodide fluorescence peak height.

**Figure 6 molecules-26-00387-f006:**
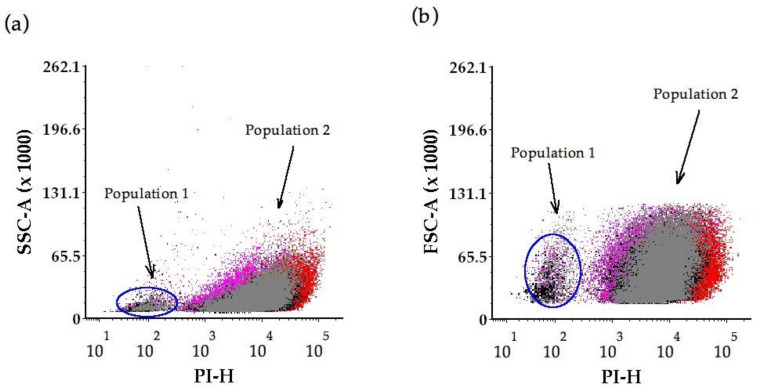
Bimodal distribution of PI fluorescence depicted for *S. cerevisiae* cells at time 0 of flow cytometry physical parameters (**a**) side scatter (SSC-A) parameter, and (**b**) forward scatter (FSC-A) parameter. Dots represent yeast replicates from the control (magenta); ultrasound treatment (black), β-glucanase enzyme treatment (grey) and microwave treatment (red).

**Figure 7 molecules-26-00387-f007:**
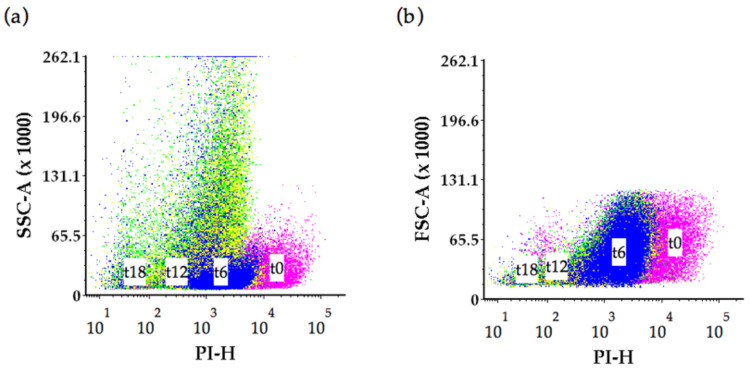
Flow cytometry (**a**) side scatter (SSC-A) parameter and (**b**) forward scatter (FSC-A) parameter, for *S. cerevisiae* cells from the control tracked over 0-, 6-, 12-, and 18-months wine ageing. Dots represent yeast replicates at 0 months (magenta); 6 month old lees (blue), 12 month old lees (yellow) and 18 month old lees (green).

**Figure 8 molecules-26-00387-f008:**
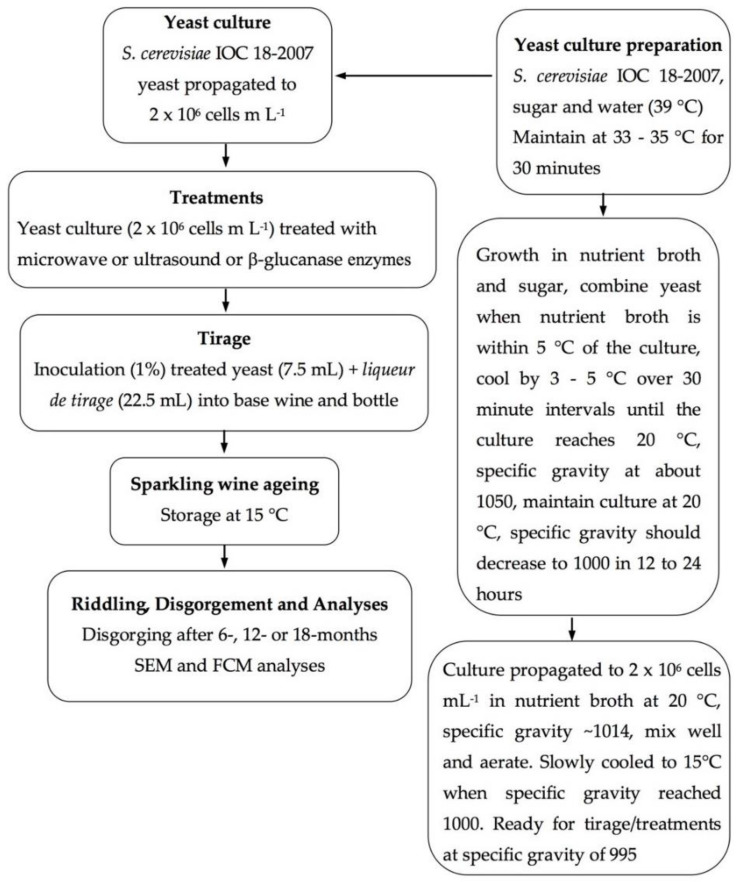
Flow diagram of sparkling wine production using yeast treatments at the tirage stage.

**Table 1 molecules-26-00387-t001:** Description of categorisation of whole-cell features of *S. cerevisiae.*

Category	Description
Smooth	Whole-cells with smooth external surfaces, mother-daughter cells, and budding, generally 1 to 5 μm size diameter
Cavitated	Modification of the cell surface to varying degrees creating an uneven cell surface
Flat	Yeast cells positioned flat on the SEM sample holder. Such cells were not classified into the ‘smooth’ or ‘cavitated’ or ‘doughnut’-shaped categories
Pitted/Porous	Indentations in the cell surface giving an impression of holes in the surface
Doughnut-shaped	Whole-cells where the centre appears to have been removed, a cell shaped like a doughnut or bagel
Fragments	Irregular small features thought to result from breakages or remains of cells that are smaller than whole-cells, generally less than 1 μm size diameter

## Data Availability

The data presented in this study are available on request from the corresponding author. The data are not publicly available due to the lead author (G.B.G.) currently undergoing PhD examination.

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
