# Peer review of "Novel Methods to Manipulate Autolysis in Sparkling Wine: Effects on Yeast"

_molecules, 2021, doi:10.3390/molecules26020387_

Round 1

Reviewer 1 Report

Microwave, ultrasound and addition of glucanase enzymes were used to accelerate the disruption of a commercial yeast,Saccharomyces cerevisiae IOC 18-2007 (Institut Oenologique de Champagne, Lallemand, France) used to prepare a yeast culture for the liqueur de tirage for secondary fermentation for bottle-fermented sparkling wine. Scanning electron microscopy and flow cytometry analyses were used to observe and describe yeast whole-cell anatomy, and cell integrity and structure via propidium iodide (PI) permeability after 6-, 12- and 18-months post-tirage. Treatments applied produced features on lees that were distinct from that of the untreated control yeast. The authors concluded that results indicate that microwave, ultrasound and addition of glucanase enzymes are all tools that could enable more rapid production of a higher quality sparkling wine. The study enables the categorization and quantification of yeast morphologies associated with the aging of sparkling wine.

  • The study on the morphological aspects is done very well and the work presented with accuracy. I found the work very interesting
  • In the past similar articles have been reported by many authors, starting from different techniques but with the same objectives.
  • However, the lack of correlation between the change in the morphological aspect of the cells with an aromatic and sensorial analysis of the wine produced makes the work poor from an oenological point of view. The aroma and sensory evaluation are considered crucial indicators to assess the highest quality of sparkling wines. For this the conclusions should be rewritten, in particular lines 599-601. This part should be justified in the text, even in the discussion
  • Another critical point is the use of a single strain. In fact Perpetuini et al. (2016, Yeast 10.1002/yea.3151) reported a high biodiversity of autolytic ability in flocculent S. cerevisiae strains. Single- strain autolytic capacity should be reported in the introduction and discussed and / or used in the conclusion.
  • Line 49 producing ethanol (about 10 % v/v), please modify, yeast appears to produce 10% ethanol
  • Lines 52-53-54 Please update some references
  • The introduction should be shortened and modified, focusing more on morphological aspects than physiological one.

Reviewer 2 Report

385 YOU CAN REFER propidium iodide fluorescence AS PIF. 

468 provide information for enzyme activity (PGNU/g)

474 Meaning of TA? Total acidity? Give informations about present (or no) of sulfites. 

Your work is well organized and described and should be accepted in present form. 

Reviewer 3 Report

The paper contains some important observations regarding morphological modifications of yeasts during sparkling wine production and autolysis, especially when autolysis is sped-up by several physical or chemical methods. The paper deserves to be published, but in the present form it is too long and contains some redundant aspects which may be better presented in a more concise manner.  Attention should also be paid to the structure and details, as explained as follows:

The chapter “Introduction” contains important elements describing the requirement of yeast autolysis for sparkling wines. However, because the paper does not present any chemical results regarding changes in the wine composition after various methods applied to speed-up the autolysis, the information about the compounds extracted should be condensed a bit more.  

The chapter “Material and Methods” is clear and contains the specific information, but it is unusually placed in the end of the paper, just before “Conclusions”. Without reading first about the treatments performed and the equipment used for analyses, it is difficult for the reader to understand the results and discussions. Thus, it is recommended to place the “Material and Methods” after “Introduction” and renumber the other chapters and sub-chapters.

The chapter “Results” is very detailed, but it is too long and difficult to follow. Same is true for the Chapter “Discussions”. Because the results are rather diverse and depend on the method of the analysis, to better explain the relevance of each type of result, before the readers forget the complex data presented, it would be better to combine results and discussion into a single chapter.

Also, in “Discussions”, some of the ideas from the introduction are repeated. Some, are not even strongly correlated with the results of this paper. For example, we find “Carew et al. (2014) found that microwave maceration was effective for rapid extraction of phenolic compounds from Pinot noir grape solids into juice” in “Discussion”, Lines 349-350, and “Microwave maceration was found effective for rapid extraction of phenolic compounds from Pinot noir grape solids into juice (Carew et al. 2014; Liu et al. 2016)” in “Introduction”, lines 103-104. Moreover, there is no strong connection between the rapid extraction of phenolic compounds by microwave maceration and the results about yeast morphological features, where it is discussed (sub-chapter “Effects of novel treatments on yeast morphological features”).

Conclusions are too brief and not entirely supported by the results presented. The fact that the treatments used affect the yeast viability and may lead to higher transfer of compounds from yeast into the wines, is not per se, in the absence of sensory or other chemical analyses, a proof that higher quality sparkling wines would be produced. (“These results indicate that microwave, ultrasound or -glucanase enzymes are tools that could enable more rapid production of a higher quality sparkling wine.” - lines 599-601).

Actually, it would be also interesting to explain if the treatments to disrupt yeast cell integrity before starting the second fermentation of sparkling wines, can lead to any measurable effect on fermentation and composition, induced by the lower number of viable cells in the tirage solution.

The Chapter “References” should be checked to see if all the references cited in the text are included. For example, the reference to “Garcia Martin et al, 2013” appears several times in the paper (lines 312, 350, 369, 371), but it is not included in the “Bibliography” section.

Other minor observations:

In line 209 it is mentioned that there is also and Appendix A and a Table A3, which are not available. And why Table A3? Is there an A1 and A2 in that appendix?

In lines 135 and 308 “(Piton, Charpentier and Troton 1988)” should be cited as Piton et al, 1998.

Round 2

Reviewer 3 Report

The revision 2 provided by the authors clarifies several aspects related to the form and background of the paper. Some of the suggested corrections have been implemented. The manuscript is now improved and warrants publication in Molecules.